# Association between triglyceride-glucose derived indices with cardiometabolic multimorbidity: Findings from the Atherosclerosis Risk in Communities study

Yi Liu[1,2,3], Dongze Li[1,2,3], Jing Yu[1,2,3], Yongli Gao[1,2,3], Wei Zhang[1,2,3], Menglin Tang[4]*

**1** West China School of Nursing, Sichuan University/Department of Emergency Medicine, West China Hospital, Sichuan University, Chengdu, Sichuan, China, **2** Institute of Disaster Medicine, Sichuan University, Chengdu, Sichuan, China, **3** Nursing Key Laboratory of Sichuan Province, Chengdu, Sichuan, China, **4** West China School of Nursing, Sichuan University/Department of Cardiac Surgery, West China Hospital, Sichuan University, Chengdu, Sichuan, China

\* menglin_tang@163.com

## Abstract

### Background

Cardiometabolic multimorbidity (CMM), characterized by the co-occurrence of diabetes mellitus (DM), stroke, and coronary heart disease (CHD), imposes substantial global health burden owing to its association with elevated mortality risk, reduced functional capacity, and increased healthcare costs. Despite its clinical importance, the value of insulin resistance (IR) and its surrogates, particularly triglyceride-glucose (TyG) indices combined with anthropometric measures, in predicting CMM remains underexplored.

### Methods

In this study, we aimed to quantify the association between TyG-derived indices and incident CMM. For this purpose, we conducted a multivariate logistic regression analysis of data of the Atherosclerosis Risk in Communities (ARIC) study, deriving adjusted odds ratios (ORs) and 95% confidence intervals (CIs). Nonlinear associations were investigated using restricted cubic spline modeling and diagnostic accuracy was evaluated using area under the curve (AUC) values from receiver operating characteristic (ROC) analyses.

### Results

Nonlinear relationships were observed between TyG-body mass index (TyG-BMI) and TyG-waist-to-height ratio, whereas TyG and TyG-waist circumference exhibited linear trends. TyG-BMI demonstrated the strongest association with CMM risk, showing a 1.61-fold increase per standard deviation (adjusted OR: 1.61; 95% CI: 1.48–1.73) and a 5.67-fold higher risk in the highest versus the lowest quartiles. Predictive performance analysis revealed that TyG-BMI was the most discriminative marker (AUC: 0.684; 95% CI: 0.664–0.705).

**Data availability statement:** The data of the ARIC studies is not freely available as it would compromise the privacy of the studies' participants, particularly since the data includes sensitive health information. Data can be requested through the Biologic Specimen and Data Repository Information Coordinating Center (BioLINCC) website (https://biolincc. nhlbi.nih.gov/studies/aric/) after creating an account and registering with the site. The data dictionary is available on this website. More information about the ARIC study can be found at https://aric.cscc.unc.edu/aric9/.

**Funding:** The author(s) received no specific funding for this work.

**Competing interests:** The authors have declared that no competing interests exist.

## Conclusions

TyG-BMI emerged as a robust predictor of CMM risk, highlighting the synergistic effects of IR and adiposity. The nonlinear risk escalation suggests threshold-dependent mechanisms, emphasizing its utility in early risk stratification.

## Introduction

Cardiometabolic multimorbidity (CMM) is the co-occurrence of two or more cardiometabolic diseases [1], including diabetes mellitus (DM), stroke, and coronary heart disease (CHD). It is widely reported to be associated with exponentially elevated mortality risk, diminished functional capacity, and increased healthcare costs [1–4]. Driven by extended life expectancies, the increasing prevalence of CMM poses substantial challenges to affected individuals and healthcare systems [5]. Previous research has predominantly focused on individual cardiometabolic conditions, with limited exploration of complex multimorbidity patterns [6–8]; consequently, the risk factors and underlying mechanisms of CMM remain poorly characterized.

Insulin resistance (IR), a physiological state marked by reduced responsiveness to insulin, plays a pivotal role in the initiation, progression, and pathogenesis of cardiometabolic diseases [9,10]. Despite the clinical relevance of IR, the complexity and cost of direct IR measurements hinder its applicability in large-scale clinical and epidemiological studies [11]. Emerging evidence suggests that triglyceride-glucose (TyG)-related indices may serve as practical surrogate markers of IR in routine clinical practice [12–14]. These indices have been associated with mortality risk, cardiovascular events (e.g., myocardial infarction and stroke), and metabolic disorders across diverse populations [15–17]. However, conclusive evidence regarding their comparative utility in predicting the incidence of CMM remains scarce.

To address the above-mentioned gaps, we integrated TyG with anthropometric measures (body weight, waist circumference (WC), and height) to construct four clinically accessible IR surrogates: TyG, TyG-body mass index (TyG-BMI), TyG-WC, and TyG-waist-to-height ratio (TyG-WHtR). Our objective was to investigate the association of TyG, TyG-BMI, TyG-WC, and TyG-WHtR with incident CMM; identify the optimal predictive biomarker for this complex multimorbidity phenotype; and offer actionable strategies for early CMM risk stratification.

## Materials and methods

### Study design and population

The Atherosclerosis Risk in Communities (ARIC) study [18], a longitudinal cohort investigation, was designed to evaluate the risk determinants of atherosclerotic progression and associated cardiovascular pathologies. Between 1987 and 1989, 15,792 participants aged 45–65 years were enrolled from four distinct U.S. communities: northwestern suburban Minneapolis (Minnesota), Washington County (Maryland), Forsyth County (North Carolina), and Jackson (Mississippi). Comprehensive baseline evaluations (designated visit 1) included a systematic collection of data

related to sociodemographic characteristics, medical histories, and clinical parameters. Subsequent follow-up evaluations were conducted at four intervals: visits 2 (1990–1992), 3 (1993–1995), 4 (1996–1998), and 5 (2011–2013). This study received ethical approval from the institutional review boards of all participating centers, and written informed consent was obtained from all enrolled individuals prior to their participation. For the present study, we obtained data from the West China Hospital of Sichuan University. The de-identified dataset was accessed in February 2024. As we used data from publicly available databases, which do not contain any identifiable information of individuals, obtaining ethical approval and patient consent was not necessary.

This analytical cohort utilized visit 1 as the baseline reference, incorporating all 15,792 initially enrolled subjects. Individuals with pre-existing CHD, stroke, or DM (n = 2,799) and those with incomplete covariate data (n = 543) were excluded from the analysis. Thus, the final analytical cohort consisted of 12,450 participants (Fig 1).

## Ascertainment of CMM

The primary outcome was the incidence of CMM, defined as the concurrent diagnosis of two or more chronic conditions, including DM, stroke, and CHD. CHD and stroke events were identified by following a standardized protocol that involved annual telephone interviews with participants or proxies, structured study visits, a systematic review of cardiovascular disease discharge records from local hospitals, and mortality certificate analyses. Diabetes was ascertained based on three complementary criteria: (1) self-reported physician diagnosis, (2) documented use of glucose-lowering medications, and (3) laboratory confirmation of glycated hemoglobin (HbA1c) level ≥ 6.5% measured during at least two separate clinical encounters.

## Assessment of TyG-related obesity indices

Blood samples were collected after an 8-h fasting period and analyzed in centralized clinical chemistry and lipid laboratories. Anthropometric measurements, including body weight, height, and WC, were obtained during physical examinations conducted at mobile examination centers. Participants were stratified into quartiles (Q1–Q4) based on the TyG, TyG-waist, TyG-WHtR, and TyG-BMI indices. Q1 served as the reference group for comparative analyses.

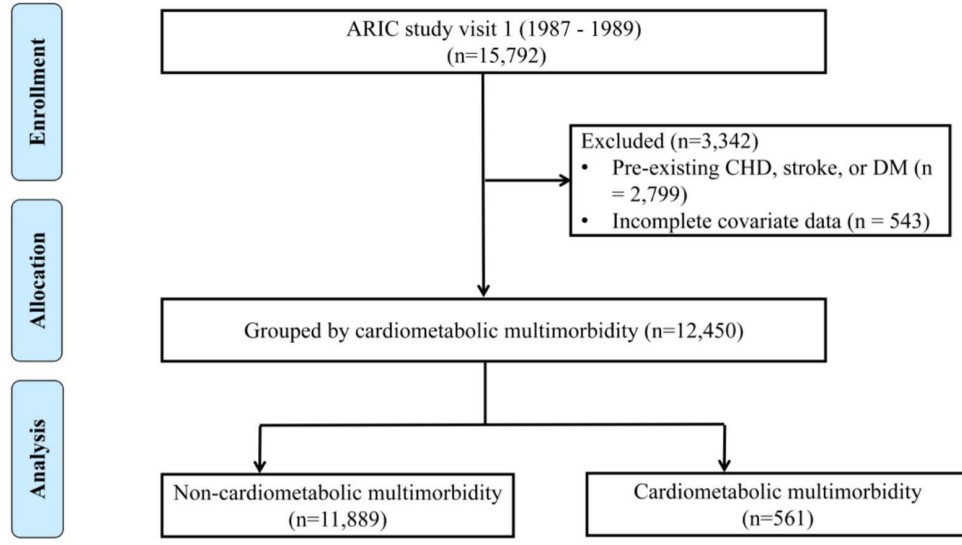

**Fig 1. Study flow chart.**

TyG, TyG-BMI, TyG-waist, and TyG-WHtR were calculated using the following formulas: (1) $TyG = \ln$ [triglycerides (mg/dL) × glucose (mg/dL)/2]; (2) $TyG\text{-}BMI = TyG \times$ body mass (kg)/height$^2$ (m$^2$); (3) $TyG\text{-}WHtR = TyG \times WC$/height; (4) $TyG\text{-}waist = TyG \times WC$.

## Covariate assessment

Baseline characteristics, including demographic variables, clinical parameters, and comorbidities, were systematically documented. Smoking status was defined as never smoked or smoked. Lipid profiles were quantified using validated methodologies. Low-density lipoprotein cholesterol (LDL-C) concentrations were calculated using the Friedewald equation, and high-density lipoprotein cholesterol (HDL-C) measurements were obtained using enzymatic colorimetric techniques following established protocols. Hypertensive status was operationally defined by meeting any of the following three criteria: (1) systolic blood pressure ≥ 140 mmHg on standardized measurement, (2) diastolic blood pressure ≥ 90 mmHg, or (3) self-reported antihypertensive medication use within the past 14-day period.

## Statistical analysis

Continuous variables were presented as either median (interquartile range) for non-normally distributed data or as mean ± standard deviation for parametric distributions, with between-group comparisons conducted using Mann–Whitney U test, Kruskal–Wallis H test, or analysis of variance as appropriate. Categorical variables are expressed as frequency (percentage) and analyzed using the chi-square test.

Multivariate logistic regression analyses were performed to evaluate the associations between cognitive performance metrics and incident atrial fibrillation, with the results reported as odds ratios (ORs) and corresponding 95% confidence intervals (CIs). The regression models were sequentially adjusted for potential confounders across three hierarchical levels. The regression model gradually adjusted for the potential confounding factors in three levels. Model 1 did not adjust for variables. Model 2 adjusted for demographic characteristics (sex, ethnicity, and age) and history of hypertension. Model 3 further adjusted for lifestyle factors (drinking and smoking status) and clinical biomarkers (total cholesterol, HDL, LDL, and triglyceride levels) on the basis of Model 2. Parallel linear regression modeling was implemented to quantify the relationships between TyG-derived indices and CMM.

Statistical significance was established at a two-tailed p-value of 0.05. Statistical computations were performed using SPSS Statistics (version 26.0; IBM Corporation, Armonk, NY, USA) and R statistical software 4.4.2 (R Foundation for Statistical Computing, Vienna, Austria).

## Results

### Baseline characteristics

The study involved 12,450 participants with a mean age of 53.96 ± 5.73 years. Demographic data revealed the following: 5,531 male individuals (44.4%), 9,611 white individuals (77.2%), and mean BMI 27.25 ± 5.13 kg/m². Hypertension history was documented in 3,168 participants (25.4%).

The study participants were stratified into two cohorts based on CMM development status. Comparative analyses between the groups demonstrated significant disparities in baseline parameters, including age, sex distribution, BMI, ethnicity, smoking status, drinking status, history of hypertension, and lipid profiles (triglycerides, total cholesterol, HDL-C, and LDL-C levels) as detailed in Table 1.

### TyG-derived indices and CMM

Multivariate-adjusted logistic regression models were used to investigate longitudinal associations between the four TyG-derived indices and incident CMM risk. As shown in Table 2, a per standard increase in the TyG index, TyG-BMI,

**Table 1.** Characteristics of adults by cardiometabolic multimorbidity.

| Characteristic | Non-CMM | CMM | p-value | q-value[1] |
|---|---|---|---|---|
| | (n = 11,889) | (n = 561) | | |
| African Americans, n (%) | 2,670 (22%) | 169 (30%) | <0.001 | <0.001 |
| Male sex, n (%) | 5,222 (44%) | 309 (55%) | <0.001 | <0.001 |
| Age, years | 54 (49, 59) | 55 (50, 60) | <0.001 | <0.001 |
| Body mass index, kg/m$^2$ | 26.4 (23.6, 29.7) | 28.8 (26.1, 32.3) | <0.001 | <0.001 |
| Drinking, n (%) | 7,048 (59%) | 301 (54%) | 0.008 | 0.009 |
| Smoking, n (%) | 6,897 (58%) | 378 (67%) | <0.001 | <0.001 |
| Hypertension, n (%) | 2,931 (25%) | 237 (42%) | <0.001 | <0.001 |
| Blood glucose, mmol/l | 5.44 (5.11, 5.77) | 5.88 (5.50, 6.27) | <0.001 | <0.001 |
| Triglycerides, mg/dl | 1.19 (0.86, 1.66) | 1.46 (1.06, 2.05) | <0.001 | <0.001 |
| Total cholesterol, mg/dl | 5.46 (4.81, 6.15) | 5.66 (5.04, 6.34) | <0.001 | <0.001 |
| HDL, mg/dl | 1.29 (1.05, 1.62) | 1.10 (0.92, 1.37) | <0.001 | <0.001 |
| LDL, mg/dl | 3.47 (2.86, 4.15) | 3.75 (3.18, 4.36) | <0.001 | <0.001 |
| TyG | 8.55 (8.21, 8.91) | 8.84 (8.49, 9.20) | <0.001 | <0.001 |
| TyG_BMI | 226 (199, 261) | 258 (231, 290) | <0.001 | <0.001 |
| TyG_waist | 824 (744, 909) | 853 (766, 931) | <0.001 | <0.001 |
| TyG_WhtR | 4.84 (4.41, 5.37) | 5.02 (4.57, 5.55) | <0.001 | <0.001 |
| Death, n (%) | 5,062 (43%) | 305 (54%) | <0.001 | <0.001 |

[1]False discovery rate correction for multiple testing.

CMM, cardiometabolic multimorbidity; HDL, High-Density Lipoprotein; LDL, Low-Density Lipoprotein; TyG, triglyceride glucose; TyG_BMI, triglyceride glucose-body mass index;TyG_waist, triglyceride glucose-waist circumference; TyG_WHtR, triglyceride glucose-waist to height ratio.

TyG-WC, and TyG-WHtR corresponded to adjusted ORs of 1.51 (95% CI: 1.38–1.65), 1.61 (95% CI: 1.48–1.73), 1.15 (95% CI: 1.06–1.25), and 1.16 (95% CI: 1.07–1.26) (Model 3, p < 0.01).

### Nonlinear associations of TyG-related indices with CMM risk

Restricted cubic spline regression analyses were conducted to flexibly model and visualize the dose–response relationships between TyG-derived indices and the risk of new-onset CMM. TyG and TyG-waist displayed linear relationships with CMM risk (P_nonlinear = 0.531; P_nonlinear = 0.258). Conversely, TyG-BMI and TyG-WHtR displayed nonlinear relationships with CMM risk (P_nonlinear < 0.001, Fig 2).

### Predictive performance of the TyG-related indices for CMM risk

Fig 3 shows the comparative predictive performances of the four TyG-derived metabolic indices for CMM risk. TyG-BMI demonstrated the highest discriminative capacity, with an area under the curve (AUC) of 0.684 (95% CI: 0.664–0.705). This was followed by the TyG index, which yielded an AUC of 0.656 (95% CI: 0.633–0.678). The nomogram in S1 Fig depicts the predicted probability of CMM using TyG-BMI, measured on a scale of 0–130.

### Discussion

This cross-sectional study elucidated the association between TyG-derived indices (TyG, TyG-BMI, TyG-WHtR, and TyG-waist) and CMM in American adults. Our analyses revealed that incident CMM has a linear relationship with TyG level and WC, contrasting with its nonlinear positive associations with TyG-BMI and TyG-waist. To draw robust conclusions, three hierarchical regression models were constructed by sequentially adjusting for demographic covariates (age, sex, and ethnicity), clinical parameters (history of hypertension, triglyceride, low-density lipoprotein, and high-density

**Table 2. Associations between the triglyceride glucose index and triglyceride glucose-related obesity indices and the cardiometabolic multi-morbidity risks.**

| Exposure | Model 1 | p-value | Model 2 | p-value | Model 3 | p-value |
|---|---|---|---|---|---|---|
| **TyG** | | | | | | |
| | **1.73 (1.59, 1.88)** | <0.001 | **1.69 (1.55, 1.84)** | <0.001 | **1.51 (1.38, 1.65)** | <0.001 |
| Q1 | 1 | | 1 | | 1 | |
| Q2 | **1.47 (1.09, 1.99)** | 0.0118 | **1.40 (1.03, 1.89)** | 0.0299 | **1.40 (1.03, 1.90)** | 0.0305 |
| Q3 | **2.85 (2.17, 3.75)** | <0.001 | **2.61 (1.98, 3.44)** | <0.001 | **2.63 (2.00, 3.48)** | <0.001 |
| Q4 | **4.21 (3.20, 5.53)** | <0.001 | **3.60 (2.73, 4.76)** | <0.001 | **3.66 (2.75, 4.88)** | <0.001 |
| | | <0.001 | | <0.001 | | <0.001 |
| **TyG_BMI** | | | | | | |
| | **1.65 (1.53, 1.77)** | <0.001 | **1.69 (1.57, 1.81)** | <0.001 | **1.61 (1.48, 1.73)** | <0.001 |
| Q1 | 1 | | 1 | | 1 | |
| Q2 | **2.14 (1.47, 3.13)** | <0.001 | **1.92 (1.31, 2.81)** | 0.0008 | **1.91 (1.30, 2.79)** | 0.0009 |
| Q3 | **4.93 (3.49, 6.96)** | <0.001 | **4.20 (2.96, 5.94)** | <0.001 | **4.12 (2.91, 5.84)** | <0.001 |
| Q4 | **7.16 (5.10, 10.04)** | <0.001 | **6.01 (4.27, 8.47)** | <0.001 | **5.67 (4.02, 8.01)** | <0.001 |
| | | <0.001 | | 0.001 | | 0.002 |
| **TyG_waist** | | | | | | |
| | **1.20 (1.10, 1.30)** | <0.001 | **1.17 (1.08, 1.27)** | <0.001 | **1.15 (1.06, 1.25)** | <0.001 |
| Q1 | 1 | | 1 | | 1 | |
| Q2 | 1.24 (0.96, 1.61) | 0.1015 | 1.20 (0.92, 1.56) | 0.1753 | 1.18 (0.91, 1.54) | 0.2147 |
| Q3 | **1.39 (1.08, 1.80)** | 0.0114 | **1.31 (1.02, 1.70)** | 0.0379 | **1.29 (1.00, 1.68)** | 0.0497 |
| Q4 | **1.76 (1.38, 2.25)** | <0.001 | **1.61 (1.26, 2.06)** | 0.0002 | **1.59 (1.24, 2.04)** | 0.0003 |
| | | <0.001 | | <0.001 | | <0.001 |
| **TyG_WhtR** | | | | | | |
| | **1.19 (1.10, 1.29)** | <0.001 | **1.18 (1.09, 1.28)** | <0.001 | **1.16 (1.07, 1.26)** | <0.001 |
| Q1 | 1 | | 1 | | 1 | |
| Q2 | **1.32 (1.02, 1.71)** | 0.0382 | 1.26 (0.97, 1.64) | 0.0807 | 1.25 (0.96, 1.63) | 0.0959 |
| Q3 | **1.46 (1.13, 1.89)** | 0.0035 | **1.38 (1.07, 1.79)** | 0.0137 | **1.37 (1.06, 1.77)** | 0.0177 |
| Q4 | **1.80 (1.41, 2.31)** | <0.001 | **1.66 (1.29, 2.13)** | <0.001 | **1.64 (1.28, 2.11)** | <0.001 |
| | | <0.001 | | <0.001 | | <0.001 |

Model 1: Unadjusted model.

Model 2: Adjusted for age, sex, ethnicity, and history of hypertension.

Model 3: Model 2 further adjusted for smoking, drinking, triglycerides, low-density lipoprotein, and high-density lipoprotein.

TyG, triglyceride glucose; TyG_BMI, triglyceride glucose-body mass index; TyG_waist, triglyceride glucose-waist circumference; TyG_WHtR, triglyceride glucose-waist to height ratio

lipoprotein levels), and behavioral variables (smoking and alcohol consumption status). TyG-BMI demonstrated superior predictive accuracy for incident CMM compared to other indices (adjusted OR: 1.61; 95% CI: 1.48–1.73). It is noteworthy that although TyG-BMI demonstrated the strongest discriminatory ability among the evaluated indices, its AUC value was 0.684 (95% CI: 0.664–0.705), which is generally considered to represent "moderate" accuracy. Consequently, the clinical value of TyG-BMI may not lie in its potential as a diagnostic tool, but as a simple and inexpensive tool for early risk identification or risk stratification in patients with CMM.

CMM has a prevalent pattern of multimorbidity associated with substantially reduced life expectancy [1]. However, risk factors for CMM remain unclear. The TyG index [8,13,15], a validated surrogate marker for IR, has emerged as an independent predictor of cardiovascular outcomes. Data from the Prospective Urban Rural Epidemiology cohort have

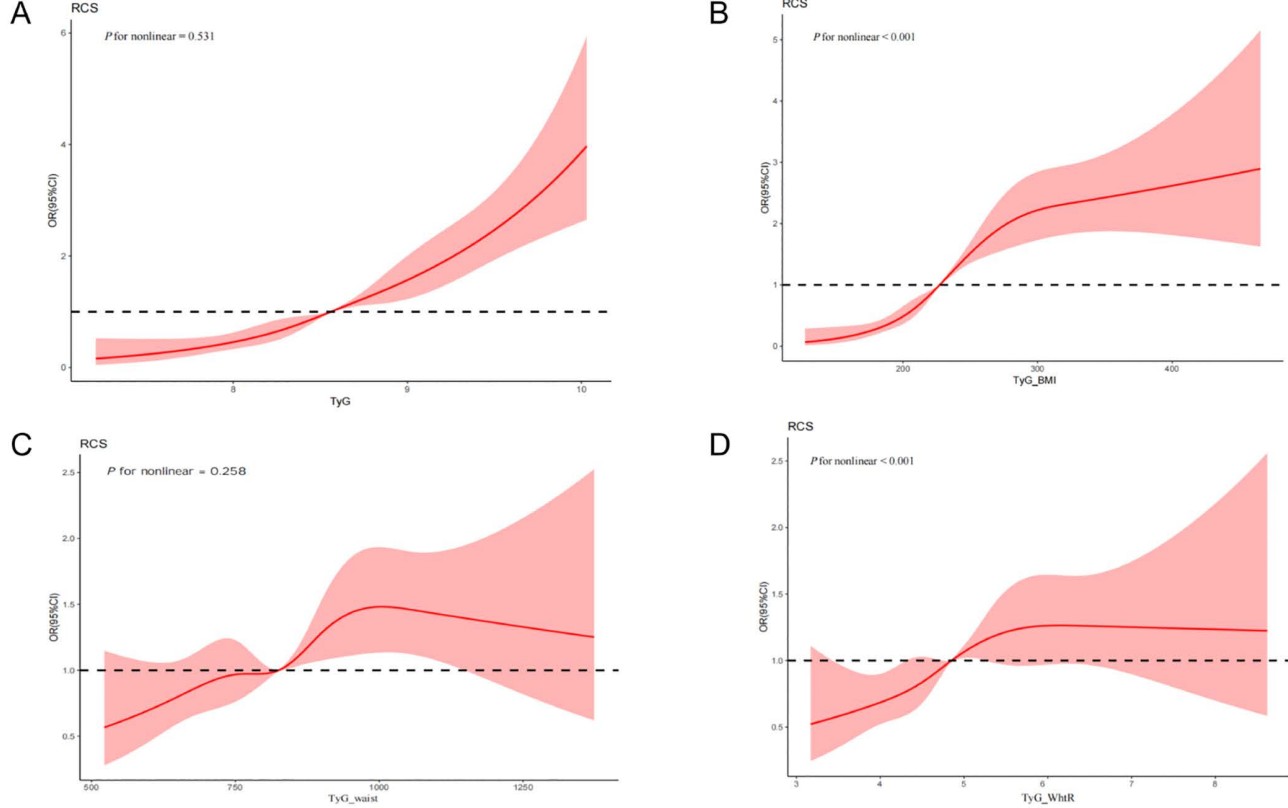

**Fig 2. Associations between (A) triglyceride-glucose (TyG) index, (B) triglyceride glucose-body mass (TyG_BMI), (C) triglyceride glucose-waist circumference (TyG_waist), and (D) triglyceride glucose waist-to-height ratio (TyG_WHtR) with the risk of cardiometabolic multimorbidity.**

established the prognostic value of TyG for cardiovascular mortality, major adverse cardiovascular events (myocardial infarction/stroke), and type 2 diabetes incidence [7]. Parallel findings from a large Korean cohort reinforce the predictive capacity of TyG for CVD risk stratification [13]. However, the association between TyG and CMM risk remains unclear.

Anthropometric indices (BMI, WC, and WHtR) serve as established proxies for obesity-mediated metabolic dysregulation [19–21]. Compelling evidence links obesity severity with a graded increase in cardiometabolic disease risk; overweight individuals demonstrate a two-time higher risk than their normal-weight counterparts, escalating to a 10-fold elevation in severe obesity [22]. Recent investigations [23–25] have consistently demonstrated enhanced CVD risk prediction when TyG is combined with obesity indices; these findings are further supported by those of a Chinese cohort study [26]. The conclusions drawn from these studies are generally consistent with our findings regarding the predictive power of TyG-derived indices for CMM risk, especially the value of TyG-BMI in predicting CMM risk.

Mechanistically, adipose tissue inflammation and mitochondrial dysfunction in obesity-induced IR may explain this association. Obese individuals with IR exhibit upregulated proinflammatory gene expression, diminished mitochondrial oxidative capacity, reduced AMP kinase activity, and elevated oxidative stress [27,28]. Notably, metabolically unhealthy normal-weight individuals paradoxically demonstrate higher cardiometabolic risks than their metabolically healthy obese counterparts, underscoring the complex interplay between adiposity distribution and metabolic homeostasis [29]. Our findings support the use of TyG together with obesity indices as an integrated biomarker for CMM risk, although mechanistic studies are needed to clarify the underlying pathways.

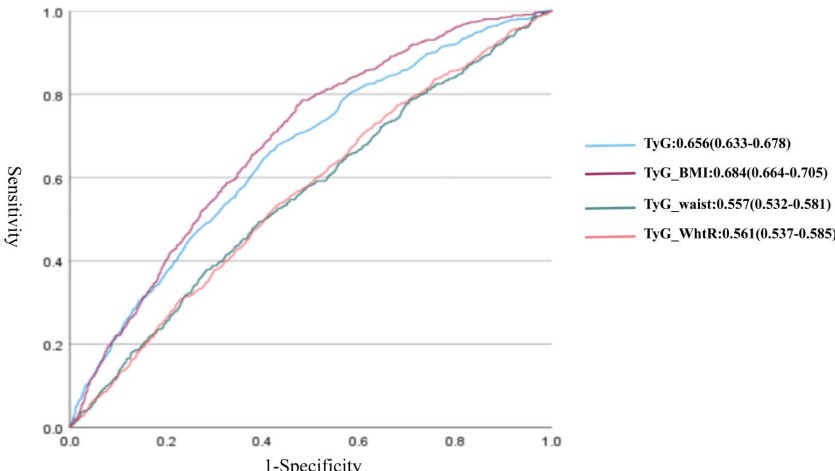

**Fig 3. Receiver operating characteristic curves and the area under the curves of triglyceride-glucose (TyG) index, triglyceride glucose-body mass (TyG_BMI), triglyceride glucose-waist circumference (TyG_waist), and triglyceride glucose-waist-to-height ratio (TyG_WHtR) in cardiometabolic multimorbidity.**

The following limitations of this study warrant further investigation. First, residual confounding factors persisted despite comprehensive adjustments for covariates. Second, the reliance on self-reported physician diagnosis of CMM introduces a potential bias in CMM classification. Third, the US community-based cohort limits generalizability to other ethnicities/ nationalities. Future multinational cohort studies employing standardized diagnostic criteria are needed to validate these findings.

## Conclusion

This study established that TyG-BMI is related to the development of CMM. The integrative nature of TyG, combined with obesity indices, captures synergistic metabolic dysregulation pathways, potentially highlighting targeted preventive strategies. Subsequent research should focus on elucidating the molecular interplay among adipose dysfunction, IR, and multimorbidity.

## Supporting information

**S1 Fig. Nomogram for risk of cardiometabolic multimorbidity(CMM) in community population.**
(TIFF)

## Acknowledgments

The authors appreciate the participation and contribution of the ARIC study participants to this research.

## Author contributions

**Conceptualization:** Dongze Li.

**Data curation:** Yongli Gao.

**Formal analysis:** Yi Liu.

**Methodology:** Yi Liu.

**Project administration:** Wei Zhang.

**Software:** Yi Liu.

**Supervision:** Yi Liu, Jing Yu.

**Visualization:** Yi Liu.

**Writing – original draft:** Yi Liu.

**Writing – review & editing:** Menglin Tang.

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
