## [Decision Letter · Decision Letter 0]

17 Nov 2025

Dear Dr. Liu,

Thank you for submitting your manuscript to PLOS ONE. After careful consideration, we feel that it has merit but does not fully meet PLOS ONE’s publication criteria as it currently stands. Therefore, we invite you to submit a revised version of the manuscript that addresses the points raised during the review process.

We look forward to receiving your revised manuscript.

Kind regards,

Alessandro Cannavo

Academic Editor

PLOS ONE

Journal Requirements:

3. Please ensure that you refer to Figure 1 in your text as, if accepted, production will need this reference to link the reader to the figure.

Reviewer's Responses to Questions

**Comments to the Author**

1. Is the manuscript technically sound, and do the data support the conclusions?

Reviewer #1: Yes

Reviewer #2: Yes

2. Has the statistical analysis been performed appropriately and rigorously?

Reviewer #1: Yes

Reviewer #2: Yes

3. Have the authors made all data underlying the findings in their manuscript fully available?

Reviewer #1: Yes

Reviewer #2: Yes

4. Is the manuscript presented in an intelligible fashion and written in standard English?

Reviewer #1: Yes

Reviewer #2: Yes

Reviewer #1: Cardiometabolic multimorbidity (CMM) represents a major public health challenge in the context of an aging population. This study moves beyond the conventional single-disease research framework by focusing on the complex co-occurrence of diabetes mellitus (DM), coronary heart disease (CHD), and stroke, thereby aligning with contemporary advances in precision prevention and integrative medicine. Drawing on the well-established Atherosclerosis Risk in Communities (ARIC) cohort, the research benefits from a large sample size (n = 12,450), an extended follow-up period, and high-quality longitudinal data, all of which contribute to the robustness and scientific validity of the findings. The article features a clear organizational structure, a sound study design, rigorous statistical methodology, and conclusions that carry meaningful clinical implications.However, the following two suggestions are proposed to further strengthen the manuscript:

1. Objective interpretation of predictive efficacy: The article identifies TyG-BMI as the indicator with the strongest discriminatory ability (AUC = 0.684; 95% CI: 0.664–0.705). While this represents the highest performance among the compared indices, the AUC value remains within the "moderate" range of discrimination, as an AUC above 0.7 is typically considered indicative of good discriminatory power. Therefore, relying solely on TyG-BMI for clinical screening may be suboptimal. It is recommended that the discussion section provide a more balanced assessment of its clinical utility, positioning TyG-BMI as a potential tool for initial screening or risk stratification rather than as an independent diagnostic criterion, thereby avoiding potential overestimation of its applicability in clinical practice.

2. Enhancement of clinical operability: To improve the translational relevance of the findings, it is suggested that a nomogram incorporating TyG-BMI and other significant predictors be included in the supplementary materials. Such a tool would enable clinicians to perform individualized prediction of CMM risk in a more intuitive and practical manner, thereby increasing the clinical applicability and impact of the study.

Reviewer #2: 1. The manuscription is written fairly well but can benefit from a final language editing.

2. The authors have clarified the data presented and how it was analysed.

3. The objective should be specific and state the actual variables tested for associations (please replace "their" with the actual variable of interest, e.g : the association of TyG, TyG-BMI, TyG-WC, and TyG-WHtR with......

4. Authors should include the models highlighted in Table 2, in the methodology first.

**Do you want your identity to be public for this peer review?** For information about this choice, including consent withdrawal, please see our Privacy Policy

Reviewer #1: No

Reviewer #2: **Yes: ** Prof. Nombeko Mshunqane

---

## [Author Response · Author response to Decision Letter 1]

20 Nov 2025

November _20, 2025

Emily Chenette

Editor-in-Chief

PLOS ONE

Manuscript ID: PONE-D-25-21918

Manuscript title: Association between triglyceride-glucose–derived indices with cardiometabolic multimorbidity: Findings from the Atherosclerosis Risk in Communities study

Dear editor:

We thank you and the reviewers for reviewing our manuscript and for providing valuable comments. We have revised the manuscript according to the comments. Our responses to the comments are provided below.

We look forward to working with you and the reviewers to move this manuscript closer to publication

Thank you for your consideration. We look forward to hearing from you.

Sincerely,

Prof. Menglin Tang, BS

Department of Cardiac Surgery, West China Hospital, Sichuan University, 37 Guoxue Road, Chengdu 610041, Sichuan, China.

Tel: +86-28-85422461, Fax: +86-28-85422288

E-mail: menglin_tang@163.com

Journal Requirements:

Thanks for your comment. The manuscript has been formatted per PLOS ONE style requirements.

2. Thank you for uploading your study's underlying data set. Unfortunately, the repository you have noted in your Data Availability statement does not qualify as an acceptable data repository according to PLOS's standards. At this time, please upload the minimal data set necessary to replicate your study's findings to a stable, public repository (such as figshare or Dryad) and provide us with the relevant URLs, DOIs, or accession numbers that may be used to access these data. For a list of recommended repositories and additional information on PLOS standards for data deposition, please see https://journals.plos.org/plosone/s/recommended-repositories.

The data from the ARIC study are not freely available as it would compromise the privacy of the study participants, particularly because the data include sensitive health information. Data can be requested through the Biologic Specimen and Data Repository Information Coordinating Center (BioLINCC) website (https://biolincc.nhlbi.nih.gov/studies/aric/) after creating an account and registering at the site. The data dictionary is available on this website. More information about the ARIC study can be found at https://aric.cscc.unc.edu/aric9/.

3. Please ensure that you refer to Figure 1 in your text as, if accepted, production will need this reference to link the reader to the figure.

We thank you for pointing this out. We have cited Figure 1 in the subsection “Study design and population” in page 5.

This analytical cohort utilized visit 1 as the baseline reference, incorporating all 15,792 initially enrolled subjects. Individuals with pre-existing CHD, stroke, or DM (n = 2,799) and those with incomplete covariate data (n = 543) were excluded from the analysis. Thus, the final analytical cohort consisted of 12,450 participants (Fig 1).

Thanks for your reminder. The reviewers' comments do not include such suggestions.

5.Please review your reference list to ensure that it is complete and correct. If you have cited papers that have been retracted, please include the rationale for doing so in the manuscript text, or remove these references and replace them with relevant current references. Any changes to the reference list should be mentioned in the rebuttal letter that accompanies your revised manuscript. If you need to cite a retracted article, indicate the article’s retracted status in the References list and also include a citation and full reference for the retraction notice.

We thank you for this important reminder. We have thoroughly reviewed the reference list and confirmed its completeness and accuracy. All references have been checked against their original sources and any formatting inconsistencies have been corrected to ensure full compliance with the journal's style guide. We confirm that none of the cited papers have been retracted.

Reviewer #1: Cardiometabolic multimorbidity (CMM) represents a major public health challenge in the context of an aging population. This study moves beyond the conventional single-disease research framework by focusing on the complex co-occurrence of diabetes mellitus (DM), coronary heart disease (CHD), and stroke, thereby aligning with contemporary advances in precision prevention and integrative medicine. Drawing on the well-established Atherosclerosis Risk in Communities (ARIC) cohort, the research benefits from a large sample size (n = 12,450), an extended follow-up period, and high-quality longitudinal data, all of which contribute to the robustness and scientific validity of the findings. The article features a clear organizational structure, a sound study design, rigorous statistical methodology, and conclusions that carry meaningful clinical implications.However, the following two suggestions are proposed to further strengthen the manuscript:

1.Objective interpretation of predictive efficacy: The article identifies TyG-BMI as the indicator with the strongest discriminatory ability (AUC = 0.684; 95% CI: 0.664–0.705). While this represents the highest performance among the compared indices, the AUC value remains within the "moderate" range of discrimination, as an AUC above 0.7 is typically considered indicative of good discriminatory power. Therefore, relying solely on TyG-BMI for clinical screening may be suboptimal. It is recommended that the discussion section provide a more balanced assessment of its clinical utility, positioning TyG-BMI as a potential tool for initial screening or risk stratification rather than as an independent diagnostic criterion, thereby avoiding potential overestimation of its applicability in clinical practice.

We thank you for this insightful comment and agree that there is a need for a balanced interpretation of the predictive efficacy of TyG-BMI. As you pointed out, an AUC of 0.684 falls within the “moderate” range of discrimination. Following the suggestion, we have revised the Discussion as follows:

It is noteworthy that although TyG-BMI demonstrated the strongest discriminatory ability among the evaluated indices, its AUC value was 0.684 (95% CI: 0.664–0.705), which is generally considered to represent "moderate" accuracy. Consequently, the clinical value of TyG-BMI may not lie in its potential as a diagnostic tool, but as a simple and inexpensive tool for early risk identification or risk stratification in patients with CMM.

2.Enhancement of clinical operability: To improve the translational relevance of the findings, it is suggested that a nomogram incorporating TyG-BMI and other significant predictors be included in the supplementary materials. Such a tool would enable clinicians to perform individualized prediction of CMM risk in a more intuitive and practical manner, thereby increasing the clinical applicability and impact of the study.

We thank you for this excellent suggestion to enhance the clinical applicability of our findings. Accordingly, we have developed a clinically applicable nomogram that integrates TyG-BMI with other significant independent predictors identified in our multivariate analysis for the individualized prediction of CMM risk. This nomogram has been included as Supplementary Figure S1 and the corresponding description has been provided in the Results. We believe that this visual tool will greatly facilitate the translation of our research into clinical practice by allowing for intuitive risk estimation.

The nomogram in S1 Fig depicts the predicted probability of CMM using TyG-BMI, measured on a scale of 0 to 130.

Reviewer #2:

1.The manuscription is written fairly well but can benefit from a final language editing.

Thank you for your comment. The revised manuscript has be checked by a Language Editing service provider.

2.The objective should be specific and state the actual variables tested for associations (please replace "their" with the actual variable of interest, e.g : the association of TyG, TyG-BMI, TyG-WC, and TyG-WHtR with......

Thank you for your careful review. We have revised the sentence as follows:

Our objective was to investigate the association of TyG, TyG-BMI, TyG-WC, and TyG-WHtR with incident CMM; identify the optimal predictive biomarker for this complex multimorbidity phenotype; and offer actionable strategies for early CMM risk stratification.

3. Authors should include the models highlighted in Table 2, in the methodology first

We thank you for this suggestion. We have revised the subsection Statistical analysis to include a detailed description of the models highlighted in Table 2. This revision provides the necessary theoretical foundation before the results are presented in the table.

The regression model gradually adjusted for the potential confounding factors in three levels. Model 1 did not adjust for variables. Model 2 adjusted for demographic characteristics (sex, ethnicity, and age) and history of hypertension. Model 3 further adjusted for lifestyle factors (drinking and smoking status) and clinical biomarkers (total cholesterol, HDL, LDL, and triglyceride levels) on the basis of Model 2.

---

## [Decision Letter · Decision Letter 1]

9 Dec 2025

Association between triglyceride-glucose derived indices with cardiometabolic multimorbidity: Findings from the Atherosclerosis Risk in Communities study

PONE-D-25-21918R1

Dear Dr. Liu,

We’re pleased to inform you that your manuscript has been judged scientifically suitable for publication and will be formally accepted for publication once it meets all outstanding technical requirements.

Kind regards,

Alessandro Cannavo

Academic Editor

PLOS One

Reviewers' comments:

Reviewer's Responses to Questions

**Comments to the Author**

Reviewer #1: (No Response)

Reviewer #2: All comments have been addressed

2. Is the manuscript technically sound, and do the data support the conclusions?

Reviewer #1: (No Response)

Reviewer #2: Yes

3. Has the statistical analysis been performed appropriately and rigorously?

Reviewer #1: (No Response)

Reviewer #2: Yes

4. Have the authors made all data underlying the findings in their manuscript fully available?

Reviewer #1: (No Response)

Reviewer #2: Yes

5. Is the manuscript presented in an intelligible fashion and written in standard English?

Reviewer #1: (No Response)

Reviewer #2: Yes

Reviewer #1: (No Response)

Reviewer #2: The authors have addressed all reviewer's comments and those for reviewer 1. Where they did not make corrections as recommended by reviewers, a detailed reason is supplied.

I Accept the manuscript for publication.

**Do you want your identity to be public for this peer review?** For information about this choice, including consent withdrawal, please see our Privacy Policy

Reviewer #1: No

Reviewer #2: No

---

## [Editor Report · Acceptance letter]

PONE-D-25-21918R1

PLOS One

Dear Dr. Liu,

I'm pleased to inform you that your manuscript has been deemed suitable for publication in PLOS One. Congratulations! Your manuscript is now being handed over to our production team.

Kind regards,

on behalf of

Dr. Alessandro Cannavo

Academic Editor

PLOS One